# Antidiarrheal Effect of Fermented Millet Bran on Diarrhea Induced by Senna Leaf in Mice

**DOI:** 10.3390/foods11142082

**Published:** 2022-07-13

**Authors:** Shujun Chen, Minquan Hao, Lizhen Zhang

**Affiliations:** College of Life Science, Shanxi University, Taiyuan 030006, China; chenshujun515@163.com (S.C.); haominquan611@163.com (M.H.)

**Keywords:** millet bran, *Bacillus natto*, fermentation, inflammatory factor, antidiarrhea

## Abstract

*Bacillus natto* is a kind of probiotic with various functional characteristics, which can produce a lot of nutrients during growth and reproduction. *Bacillus natto* was used as strain, the number of viable bacteria and the content of soluble dietary fiber in millet bran were used as indexes to study the effects of inoculum size, fermentation time, and fermentation temperature on the fermentation effect, and the optimal fermentation conditions were determined by a response surface experiment. The antidiarrhea effect of fermented millet bran prepared under the best technological conditions was evaluated. The results showed that the optimum fermentation conditions were as follows: inoculum size was 7.48%, fermentation time was 47.04 h, and fermentation temperature was 36.06 °C. Under the optimal fermentation conditions, the viable bacteria count of millet bran was 8.03 log CFU/mL and the soluble dietary fiber content was 12.14%. The fermented millet bran can significantly reduce the intestinal thrust rate and serum levels of IL-6, IL-12, and TNF-α, and significantly increase the secretion of SIgA in the intestinal mucosa, which can relieve diarrhea induced by senna leaf in mice. The results of this study can provide the scientific basis for deep processing of millet bran and efficient utilization of fermented millet bran, and also provide the theoretical basis for clinical treatment of diarrhea.

## 1. Introduction

Millet bran is a by-product of millet processing, mainly composed of millet shell, seed coat, aleurone layer, and germ, accounting for about 6–8% of the total mass of millet. In China, the annual output of millet is about 4.5 million tons, accounting for about 80% of the world’s total output. The annual production of millet bran is about 400,000 tons [1]. Millet bran is rich in vitamins, minerals, flavonoids, polyphenols, phytosterols, dietary fiber, and other biologically active ingredients [2], which makes it have the functional characteristics of lowering blood lipids, antifatigue, antibacterial, anti-inflammatory, enhancing immunity, and preventing cardiovascular and cerebrovascular diseases [3,4,5,6]. At present, millet bran is mainly made into livestock and poultry feed, its rich nutrients and bioactive components have not been fully utilized, and the resource utilization rate is low [7].

Among many food processing methods, fermentation is an effective way to significantly increase the added value of millet bran and its biological activity, and it also helps to make millet bran a functional substance with high biological activity. Studies have shown that after microbial fermentation, the content of total phenolics and total flavonoids in millet bran increased by 59.2% and 56.6%, respectively [8], the content of soluble dietary fiber (SDF) in millet bran meal increased by 8.88% [9], the content of soluble dietary fiber in millet bran reaches 13.24%, which is 5.9 times higher than that in raw materials [10], the total antioxidant capacity of millet bran fermentation extract higher than the unfermented extract [11], and the soluble dietary fiber content of millet bran after natto fermentation increased from 2.3% to 13.2% [12]. Some studies on fermented millet bran have shown that the fermentation process of millet bran adopts *Rhizopus oryzae* [13], *Rhizopus*
*oligosporus* [14], Monascus [15], yeast [16], etc. Among them, *Bacillus natto* can produce active antibacterial substances that inhibit pathogenic bacteria [17], can secrete amylase, protease [18], lipase [19], plant acidase, and other extracellular enzymes [20], and can promote intestinal microecological balance [21], regulate intestinal metabolism [22], and enhance immunity [23] and other functions. The application of *Bacillus natto* to the fermentation of millet bran by microbial fermentation can provide a theoretical basis for the development of millet bran deep processing products.

Diarrhea is one of the common diseases in gastroenterology and is a common symptom of many diseases. The main symptoms are increased defecation frequency, loose stool, and even watery diarrhea [24]. Prolonged diarrhea can cause dehydration and electrolyte imbalance in the body [25]. Diarrheal disease has become a worldwide public health problem due to its high incidence, and it is also one of the important causes of malnutrition, growth retardation, and death in children.

Previous studies have shown that fermentation of millet bran not only increases polyphenol content [26], but also increases the content of tryptophan [27], which has been shown to improve intestinal inflammation [28]. A study by Agista [29] found that fermented rice bran protects the mouse gut from dextran sodium sulfate (DSS)-induced inflammation. Kataoka [30] and other studies found that fermented brown rice inhibited the acute colitis induced by sodium glucan sulfate in rats. The addition of fermented rice bran ameliorated the clinical manifestations of DSS-induced colitis inflammation and reduced the levels of pro-inflammatory cytokines in the small and large intestines [31].

In this study, single factor and response surface methodology were used to optimize the fermentation conditions of millet bran by *Bacillus natto*. Meanwhile, the effects of fermented millet bran on serum IL-6, IL-12, and TNF-α levels, intestinal mucosal SIgA level, and intestinal propulsive rate of senna leaf induced diarrhea mice was also studied, which could offer an instructive guide for the development and application of millet bran functional food [32] and clinical treatment of diarrhea.

## 2. Materials and Methods

### 2.1. Materials and Reagents

Millet bran was obtained from Jinzhong Fengyuan planting professional cooperative (Jinzhong, Shanxi, China). Bacillus subtilis (CICC 10453) was purchased from the Chinese Strain Preservation Center. Senna leaf was purchased from Bozhou Mingtao Biology Co., LTD. (Bozhou, Anhui, China). Loperamide hydrochloride capsule (2 mg/tablet) was purchased from Xi’an Janssen Pharmaceutical Co., LTD., Batch No. KIJ09E6. (Xi’an, Shanxi, China). The assay kits for enzyme-linked immunosorbent assay (ELISA) kits for interleukin (IL)-6 and interleukin (IL)-12 as well as tumor necrosis factor (TNF)-α and secretory immunoglobulin A (SIgA) were purchased from Shanghai Aimeng Youning Biotechnology Co., LTD. (Shanghai, China). Volume fraction 95% ethanol, glucose, sodium chloride, beef extract, peptone, and AGAR were purchased from Sinopharm Chemical Reagent Co., LTD. (Shanghai, China). All reagents and chemicals are analytical grade.

### 2.2. Experimental Animals

Specific pathogen-free male KM mice (4 weeks old, (20 ± 2) g) were purchased from Beijing Weitong Lihua Laboratory Animal Technology Co., Ltd. (Beijing, China, certificate number: SCXK (Jing) 2021-0011). All mice were kept under the circumstance of 25 ± 2 °C in a 12 h light-dark cycle and were freely given access to water and standard chow. Before the formal beginning of the experiments, the mice were placed in the animal room for adaptive feeding for 7 days, during which they were free to ingest food and water.

### 2.3. Preparation of Fermented Millet Bran

#### 2.3.1. Preparation Process

Millet bran was crushed, screened through 80 mesh, petroleum ether 1:4 (g/mL) degreased, repeated three times, washed with 80% ethanol, and dried for later use [33]. Weigh 10 g defatted millet bran in a 250 mL conical flask, add 0.1 g glucose, 0.5 g sodium chloride and 100 mL water, and sterilize at 121 °C for 15 min. After cooling, the suspension of *Bacillus natto* was added to the culture medium and cultured in an incubator. After that, the fermentation liquid was centrifuged and the supernatant was freeze-dried.

#### 2.3.2. Single-Factor Experiment

Taking viable count and soluble dietary fiber (SDF) content as evaluation indexes, the basic fermentation conditions were set as follows: inoculum size was 7%, fermentation time was 48 h, fermentation temperature was 35 °C, and natural pH. Single-factor tests were designed as follows: inoculum size at 1%, 3%, 5%, 7%, and 9%, fermentation time at 12 h, 24 h, 36 h, 48 h, and 60 h, fermentation temperature at 25 °C, 30 °C, 35 °C, 40 °C, and 45 °C. The effects of millet bran fermentation were studied by using the average value from three repeated tests.

#### 2.3.3. Response Surface Design of Experiments

Based on a single factor experiment, viable count (Y_1_) and SDF content (Y_2_) were taken as the response values. According to Box–Behnken Design principle, the experimental factors and levels were designed using Design Expert 8.0 software. The effects of inoculum size, fermentation time, fermentation temperature, and the interaction of each factor on viable count and crude SDF content of fermented millet bran were investigated. The factors and levels were shown in Table 1.

#### 2.3.4. Indicator Determination

The number of viable bacteria was determined with reference to the China national food standard (GB4789.2-2016).

Crude SDF content was determined as follows [34]: after millet bran fermentation, the fermentation liquid was centrifugalized at 4000 r/min for 10 min, filtrate was collected and added with 95% ethanol (*v*/*v*) by four times the volume. Alcohol was settled overnight at 4 °C and the precipitate was centrifuged, then supernatant was discarded and the substrate was dried, which was the crude SDF. The calculation formula is as follows:(1)$$\mathrm{SDF}\;\left(\%\right)=\frac{M_{1}\;\left(g\right)}{M\;\left(g\right)}\times 100\%$$
where M_1_ is the mass of SDF and M is the mass of millet bran.

### 2.4. Antidiarrhea Experiment of Fermented Millet Bran

After adapting to the environment for one week, the mice were randomly divided by their body weight into normal control group (NC), model control group (MC), positive control group (PC), fermented millet bran group (FMB), and millet bran porridge oil group (MBPO), with twelve mice in each group. Before the experiment, the mice were fasted for 12 h and had free access to water. Except for the normal control group, which was given normal saline (0.4 mL/20 g.bw.d), the other groups were given 1 g/mL senna decoction (0.4 mL/20 g.bw.d), continuous gavage for 7 days. Each mouse began to have dull fur, loose stools, and watery stools, indicating that the modeling was successful. After the model was successfully established, the senna water decoction was administered to the stomach for the last time, and the corresponding drugs were administered to the stomach 2 h later: the fermented millet bran group was gavaged with fermented millet bran extract freeze-dried power (2.05 g/kg.bw.d), the millet bran porridge oil group was gavaged with millet bran porridge oil freeze-dried power (10.27 g/kg.bw.d), the positive control group was gavaged with 0.2 g/mL loperamide hydrochloride suspension (0.4 mL/20 g.bw.d), and the normal control group and model group were gavaged with normal saline for 7 days.

### 2.5. General Indicator Observations

The mental state and food intake of mice were observed during the experiment. During the period of intragastric administration, the body weight and daily feed surplus of the mice were weighed regularly every day, and their changes were recorded.

### 2.6. Determination of Diarrhea Index in Mice

After the establishment of the diarrhea model, the defecation was observed and recorded at an interval of 1 h. The mice were placed in different cages with white blotting paper on the bottom of the cages and a wire mesh rack on top. The mice were placed on a wire mesh rack and separated from the blotting paper to avoid paper sucking and stomping on feces. The filter paper was replaced every 1 h, and the total number of defecations, the number of loose defecations, and the grade of loose defecation in 4 h were observed and recorded [35]. Diarrhea index was determined on the 1st, 3rd, and 7th day of administration.

### 2.7. Detection of IL-6,IL-12, and TNF-α Content in the Serum of Mice

After the observation of diarrhea mice, the mice fasted for 12 h. After anesthesia, the whole blood was collected from the eyeballs and centrifuged at 12,000 r/min at 4 °C for 15 min. The serum was collected and labeled and stored at −80 °C for future use. IL-6, IL-12, and TNF-α ELISA kits were used to detect the contents of various cytokines in different groups of mice.

### 2.8. Determination of SIgA Content in Small Intestinal Mucosa

The small intestine of the proximal ileocecal part of mice was taken 10 cm aseptically, tiled and dissected longitudinally, and the feces in the intestine were removed gently. The surface of intestinal mucosal tissue was scraped lightly with a slide for 1 g and collected in a centrifugal tube. PBS of 0.01 mol/L was added for 1 mL, fully dissolved for 1 h, and centrifuged at 2000 r/min for 10 min. The content of SIgA was determined by the slgA kit [36].

### 2.9. Determination of Thymus and Spleen Index

The cervical vertebra of the mice was dislocated after blood collection, and the mice were dissected immediately. The thymus and spleen of the mice were collected, washed with normal saline, and the water on the surface was drained by filter paper, and the organ index was calculated. The viscera index was calculated as follows:(2)$$\mathrm{Viscera}\;\mathrm{index}\left(\mathrm{mg}/g\right)=\frac{\mathrm{Viscera}\;\mathrm{weight}\left(\mathrm{mg}\right)}{\mathrm{body}\;\mathrm{weight}\left(g\right)}$$

### 2.10. Determination of Small Intestine Propulsion Rate

After the last administration, mice were given 0.15 mL of 0.5% CMC-Na at 5% by intragastric administration. After 30 min, the mice were dissected, the abdominal cavity was opened, and the intestinal segment from the gastric pylorus to ileocecum nodule was removed, after which the length of segment A and total length B running at the end of carbon are measured.
(3)$$\mathrm{small}\;\mathrm{intestinal}\;\mathrm{propulsion}\;\mathrm{rate}\left(\%\right)=\frac{A}{B}\times 100\%$$
where A is the length of the end of the carbon and B is the total length of small intestine.

### 2.11. Small Intestine Tissue Section

The ileum about 3 cm near the cecum segment was removed and fixed in 10% neutral formaldehyde. After undergoing routine dehydration, a paraffin-embedded, section and was stained with hematoxylin and eosin (HE) for microscopic intestinal tissue changes observation.

### 2.12. Statistical Analysis

All data were expressed as mean ± SEM (standard error of mean) and all analyses were performed using SPSS 17.0 software (SPSSInc., Chicago, IL, USA). Differences among various groups were analyzed by one-way ANOVA and the results with *p* < 0.05 were regarded as statistically significant. The response surface optimization experiment was designed by Design-Expert 8.0.6 software (China), and the diagram was visualized by Origin 8.6 software (China).

## 3. Results

### 3.1. Results of the Single-Factor Experiment

With the increase of inoculum size, fermentation time, and fermentation temperature, the viable count and the content of soluble dietary fiber in fermented millet bran increased firstly and then decreased. The maximum values were obtained at inoculum size of 7%, fermentation time of 48 h, and fermentation temperature of 35 °C, respectively (Figure 1). When the inoculum size is too small, the number of viable bacteria in the fermentation system is less, and there is no sufficient fermentation. With the increase of inoculum size, the number of viable bacteria increased, and the SDF content also increased. When the inoculum size was too large, the viable count was too much, the decomposition rate was too fast, and the nutrient supply of the strain was insufficient, leading to the decrease of the SDF content. The reason for the decrease of SDF content after 48 h may be that the prolonged fermentation time reduces the required nutrients, accumulates a large number of harmful substances, and microorganisms can no longer survive, which affects the SDF content [9]. The reason for the decrease of viable count and SDF content may be that 35 °C is the temperature closest to the growth of the bacteria. Too low or too high temperature has certain influence on the growth and metabolism of the bacteria, thus affecting the effect of millet bran fermentation [37]. Therefore, 7% inoculum size, 48 h fermentation time, and 35 °C fermentation temperature were selected for follow-up experiments.

### 3.2. Result of Response Surface

#### 3.2.1. Response Surface Experiment Design and Results

Design-Expert software was used to perform multiple regression fitting analysis on the data in Table 2, and the quadratic multinomial regression equation of viable count and SDF content in millet bran fermented by *Bacillus natto* was obtained as follows:Y_1_ = 8.01 + 0.10A + 0.095B + 0.090C + 0.087AB − 0.083AC − 2.5 × 10^−3^BC − 0.20A^2^ − 0.19B^2^ − 0.14C^2^ (R_1_^2^ = 95.88%)
Y_2_ = 12.04 + 0.57A + 0.079B + 0.36C − 0.16AB − 0.073AC + 0.042BC − 1.01A^2^ − 0.89B^2^ − 0.91C^2^ (R_2_^2^ = 99.64%)

Analysis of variance of the regression model was summarized in Table 3. The results showed that the regression model of viable count has an acceptable *p*-value (<0.05) and R_1_^2^ value (0.9588), which can describe the viable count of the fermentation process effectively. The order of factors affecting the viable count in fermented millet bran was that inoculum size was greater than fermentation temperature than fermentation time. The regression model of soluble dietary fiber content has an acceptable *p*-value (<0.05) and R_2_^2^ value of 0.9964, which can describe the SDF content of the fermentation process effectively. The order of factors affecting the content of soluble dietary fiber in fermented millet bran is that the inoculum size is greater than the fermentation time and the fermentation temperature.

#### 3.2.2. Effects of Interaction of Three Factors on Viable Count

When the fermentation temperature was constant, the viable count increased first and then stabilized with the increase of inoculum size, and increased first and then decreased with the extension of fermentation time. The contour line tended to ellipse, indicating a significant interaction between inoculum size and fermentation time (Figure 2a). When the fermentation time was constant, the viable count increased firstly and then stabilized with the increase of inoculum size and fermentation temperature (Figure 2b). When the inoculum size was constant, the viable count increased first and then decreased with the increase of fermentation time, and increased first and then stabilized with the increase of fermentation temperature, which was consistent with the trend in the previous figure (Figure 2c). A has a greater influence on the fermentation effect than B and C.

#### 3.2.3. Effects of the Interaction of Three Factors on SDF Content

When the fermentation temperature was constant, the content of SDF increased firstly and then stabilized with the increase of inoculum size and fermentation time (Figure 3a). When the fermentation time was constant, the content of SDF increased firstly and then stabilized with the increase of inoculum size and fermentation temperature (Figure 3b). When the inoculation amount was constant, the content of SDF increased first and then decreased with the extension of fermentation time, and increased first and then stabilized with the increase of fermentation temperature (Figure 3c). A has a greater influence on the fermentation effect than B and C.

#### 3.2.4. Determination of the Optimal Fermentation Process

According to the established quadratic regression model, the optimal conditions for fermenting millet bran are as follows: inoculum size was 7.48%, fermentation time was 47.04 h, and fermentation temperature was 36.06 °C. Under these conditions, the viable count was 8.03 log CFU/mL and SDF content was 12.14%. Combined with the actual consideration, the optimal fermentation conditions were determined as follows: inoculum size was 7.5%, fermentation time was 47 h, and fermentation temperature was 36 °C. Under the above conditions, the viable count was 8.08 log CFU/mL, SDF content was 11.96%, and the relative error was less than 0.05. The regression equation has a high fitting degree and can be used to optimize the fermentation process of millet bran.

### 3.3. General Observation of Mice

The initial body weight of mice in each group was similar (Figure 4). After 4 days of modeling, the weight increase of the normal control group was higher than that of other groups. After 7 days of modeling, the weight of the normal control group was significantly higher than that of other groups (*p* < 0.05), indicating that the diarrhea model was successfully built. After successful modeling on day 7, drug administration was started. On day 10, except for the model group, the body weight of mice in other groups increased rapidly. On day 13, the body weight of each group was significantly higher than that of model group (*p* < 0.05). After the establishment of the diarrhea model, the number of defecations in mice increased, and the weight gain was slow. After the beginning of the administration, the diarrhea situation of mice improved, and the weight gain gradually became closer to the normal control group.

### 3.4. Effect of Diarrhea Index in Mice

The diarrhea index of the normal control group was 0, while the diarrhea index of the model group was significantly different from that of the normal control group (*p* < 0.05), indicating the molding success (Table 4). The diarrhea index of the fermented millet bran group was significantly lower than that of model group (*p* < 0.05), indicating that fermented millet bran can improve diarrhea caused by Senna leaf in mice [38].

### 3.5. Effects of Serum Inflammatory Factors

IL-6, IL-12, and TNF-α are all representatives of cellular inflammatory factors. When tissue cells are persecuted by inflammatory factors, the levels of IL-6, IL-12, and TNF-α in cells increase and are released into the blood, and the levels of IL-6, IL-12, and TNF-α in serum show an increasing trend. Compared with the normal control group, IL-6, IL-12, and TNF-α concentrations in model group were significantly increased, with statistical significance (*p* < 0.05). Compared with the model group, the concentrations of IL-6, IL-12, and TNF-α in the fermented millet bran group were lower than those in the model group, with a significant difference (*p* < 0.05) (Figure 5a–c). For example, fermented millet bran can significantly reduce serum pro-inflammatory cytokines TNF-α and IL-6, thus improving recurrent colitis, and has a significant protective effect on colonic inflammation [31]. Fermented rice bran supplementation is not limited to preventing inflammation, but also contributes to the intestinal repair of persistent colitis [26].

### 3.6. Effect of SlgA Content in Intestinal Mucosa

SlgA content in the model group was significantly lower than that in the normal control group (*p* < 0.05), and SlgA concentration in the fermented millet bran group was significantly higher than that in model group (*p* < 0.05) (Figure 5d). Beneficial bacteria in fermentation can prevent the atrophy of normal mucosal cells and goblet cells, protect the intestinal mucosal barrier, and ensure normal secretion of intestinal mucosal SlgA [39].

### 3.7. Thymus and Spleen Index

Compared with the normal control group, the spleen and thymus indexes of the model group were not significantly different (*p* > 0.05). The fermented millet bran group and millet bran porridge oil group had no significant difference compared with the model group (*p* > 0.05) (Figure 6). There was no significant difference in organ index in each group, which may be due to the short experimental time and insignificant damage.

### 3.8. Effect of the Small Intestine Propulsion Rate

Compared with the model group, the small intestine propulsion rate of other groups was significantly reduced (Figure 7), indicating that fermented millet bran has a certain relieving effect on diarrhea induced by Senna leaf.

### 3.9. Section Observation of Small Intestine

In the normal control group, the intestinal tissue structure was complete and clear, the mucosa was orderly arranged, and no inflammatory reaction was observed (Figure 8a). In the model group, small intestinal mucosa villi were shed, intercellular connective structure was deficient, the normal intestinal structure was lost, and inflammatory cells were infiltrated (Figure 8b). Compared with the model group, the villi of the small intestinal mucosa in the fermented millet bran group were basically not ablated and a small number of inflammatory cells were infiltrated (Figure 8d).

## 4. Discussion

The number of viable bacteria and SDF content of millet bran fermented by *Bacillus natto* under the optimal fermentation conditions were 8.03 log CFU/mL and 12.14%, respectively. The results of the antidiarrhea experiment showed that fermented millet bran could reduce the levels of IL-6, IL-12, and TNF-α in the serum of mice, and increase the level of SlgA, to alleviate diarrhea in mice.

Millet bran is rich in bioactive phytochemicals, among which soluble dietary fiber has a prebiotic function. After entering the intestine, soluble dietary fiber will be utilized by some probiotics to promote their growth, balance the intestinal flora, and nourish the intestine. During fermentation, enzymes produced by microorganisms acting as a starter can hydrolyze complex compounds in their bound form into compounds in their free form, and biological activity increases as bioactive compounds increase. Studies have shown that millet bran fermented by *Bacillus natto* can significantly increase its soluble dietary fiber content, and adding millet bran with prebiotic function is an effective probiotic nutritional approach. Several studies have shown that compared with unfermented rice bran, fermented rice bran has higher bioactive substance content and functional characteristics [40].

Currently, senna leaves are used to establish animal models to support the study of modern diarrhea-related diseases due to their diarrhea-inducing properties. In the diarrhea model caused by senna leaf decoction, its mechanism is generally believed to be that senna leaf is not absorbed by the small intestine after entering the large intestine and that some effective components of Senna leaf react in the large intestine, while some react in the small intestine. This affects the sodium and potassium pump of small intestinal epithelial cells, and in turn produces inflammatory factors to cause diarrhea. It is also believed that a variety of anthrax substances are the main cause of diarrhea caused by soap leaves, such as saponins, aloe-emodin, emodin acid, and chrysophanol. Zhang [41] showed that Senna leaves targeting CYP3A4 alter the microbial ecology of intestinal microorganisms and interfere with the tryptophan metabolism of intestinal flora, which may be one of the pharmacological mechanisms of the laxative effects of Senna leaves.

The gastrointestinal tract is not only a digestive organ, but also an important immune organ. When the immune function is abnormal, such as immune overreaction or immune tolerance, various gastrointestinal diseases will occur, which then cause an immune function imbalance in the body. Cytokines are the basic cytokines in the immune regulatory network, which participate in the growth, differentiation, and function regulation of various tissues and cells, and play a key role in immune and inflammatory responses [42].

IL-6, IL-12, and TNF-α are all representatives of cellular inflammatory factors. When tissue cells are persecuted by inflammatory factors, the levels of IL-6, IL-12, and TNF-α in cells increase and are released into the blood, and the levels of IL-6, IL-12, and TNF-α in serum show an increasing trend. When TNF-α acts on vascular endothelial cells, it increases their permeability, resulting in a large amount of exudation from the intestinal wall. Therefore, the high level of TNF-α can reflect the increase of intestinal wall permeability to a certain extent and cause diarrhea-related symptoms in severe cases [43]. IL-6 can induce the differentiation of B cells and T cells, participate in the body’s immune response, and accelerate the occurrence of inflammation [44]. IL-12 is a cytokine that can independently induce the differentiation and proliferation of Th1 cells. It can promote the degranulation of neutrophils, release elastin, damage endothelial cells, cause stagnation of microcirculation blood flow, cause tissue necrosis, and lead to organ dysfunction [45]. In this study, fermented millet bran can significantly reduce the contents of IL-6, IL-12, and TNF-α and alleviate the inflammatory response caused by Senna leaves. Previous studies have reported that mRNA levels of IL-1β, TNF-α, and IL-6 increased in mice after acute colitis [27,46]. In this study, the contents of IL-6, IL-12, and TNF-α in diarrhea mice were higher than those in normal mice, and the contents of IL-6, IL-12, and TNF-α in mice were reduced after treatment with fermented millet bran, indicating that fermented millet bran can reduce the content of pro-inflammatory factors and improve diarrhea in mice caused by Senna leaves. The ovalbumin (OVA) model showed that fermented rice bran extract reduced TNF-α, interferon (IFN-γ), IL-6, and IL-10 in mice [47]. In a previous study, supplementation of fermented rice bran decreased TNF-α, IL-6, and IL-1β levels, thereby increasing body weight and fecal concentration, and reducing intestinal bleeding caused by increased short-chain fatty acids (SCFA) and trypamine to protect the C57BL/6N mice from ulcerative colitis [48], consistent with this study.

SIgA is the most vital antibody on the mucosal surface, which can effectively prevent the adhesion of pathogens on the mucosal surface and resist the infection and invasion of pathogens. Loss of local SIgA increases susceptibility to mucosal infection. SIgA is a crucial factor in mucosal resistance to infection, and is also a significant indicator for the diagnosis and prognosis of various mucosal diseases. SIgA protects the intestinal mucosa by reducing its direct contact with pathogens [49]. SIgA is involved in the composition of the intestinal biological barrier. After binding with pathogens, SIgA can resist the invasion of pathogens by promoting intestinal peristalsis, villus movement, loose mucus layer flow, and other physical ways. It can also neutralize with enzymes and enterotoxins in the intestinal tract through chemical neutralization, which can double block the contact between pathogens and mucosa in space and time. Another important mechanism of SIgA to protect the mucosa from pathogens is to maintain the dynamic balance of the internal environment on the mucosal surface, so that the intestinal symbiotic bacteria can reach a relatively stable state [50]. This study showed that the SIgA content in the model group was lower than that in the normal group, mucosal congestion and inflammatory cell infiltration were significantly reduced after treatment with fermented millet bran, and intestinal SIgA content was significantly increased, indicating that the occurrence of diarrhea was related to the weakening of local intestinal mucosal immune function to a certain extent.

## 5. Conclusions

In conclusion, fermented millet bran alleviates diarrhea by reducing IL-6, IL-12, and TNF-α levels in serum, increasing SIgA levels in the intestinal mucosa, and improving intestinal tissue lesions. In recent years, dietary functional foods have attracted attention for their nutritional value, therapeutic value, and low side effects. The results of this study provide a theoretical basis for the development of millet bran antidiarrheal products.

## Figures and Tables

**Figure 1 foods-11-02082-f001:**
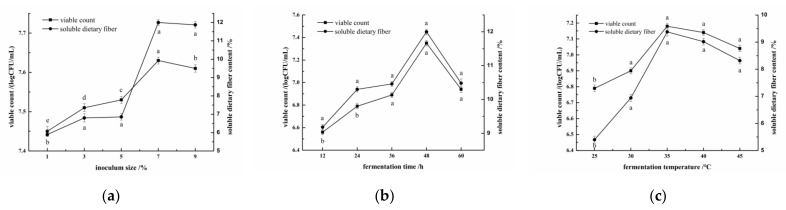
Effect of (**a**) inoculum size, (**b**) fermentation time, and (**c**) fermentation temperature on viable count and content of SDF. The letters a, b, c, and d in the figure represent significant differences between different letters. (*p* < 0.05).

**Figure 2 foods-11-02082-f002:**
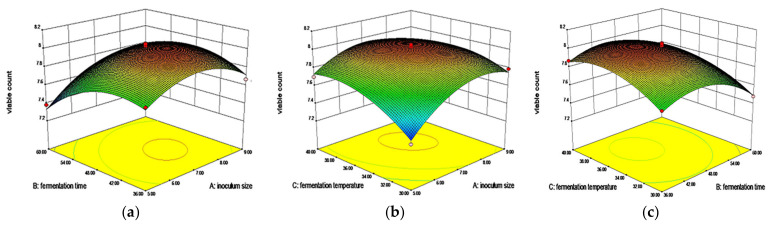
Effects of (**a**) inoculum size and fermentation time, (**b**) inoculum size and fermentation temperature, and (**c**) fermentation time and fermentation temperature on viable count.

**Figure 3 foods-11-02082-f003:**
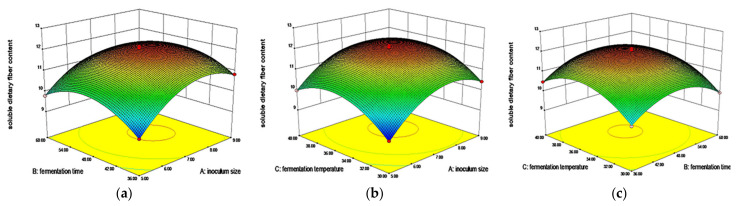
Effects of (**a**) inoculum size and fermentation time, (**b**) inoculum size and fermentation temperature, and (**c**) fermentation time and fermentation temperature on SDF content.

**Figure 4 foods-11-02082-f004:**
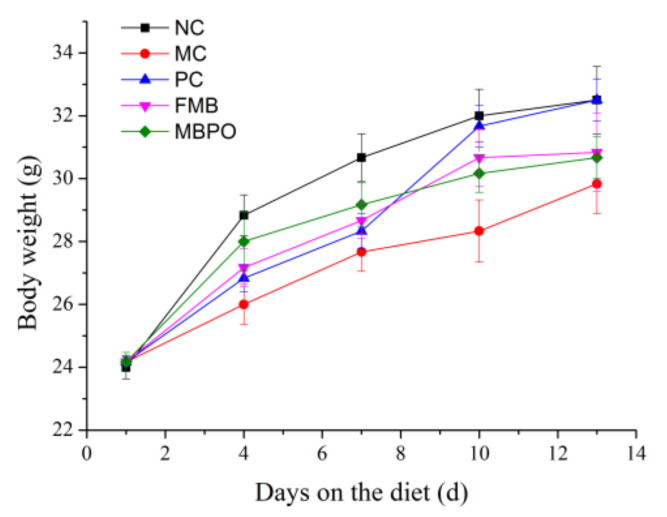
Body weight changes of mice in each group during the experiment.

**Figure 5 foods-11-02082-f005:**
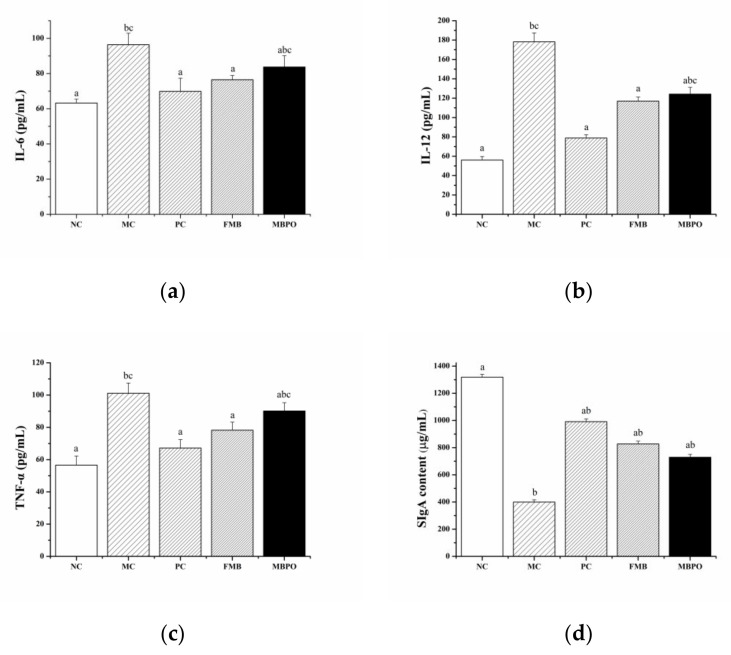
Effects of fermented millet bran on the expression of inflammatory cytokines (**a**) IL-6, (**b**) IL-12, and (**c**) TNF-α and intestinal mucosa (**d**) SIgA in mice. ^a^
*p* < 0.05 vs. MC group, ^b^
*p* < 0.05 vs. NC group, ^c^
*p* < 0.05 vs. PC group.

**Figure 6 foods-11-02082-f006:**
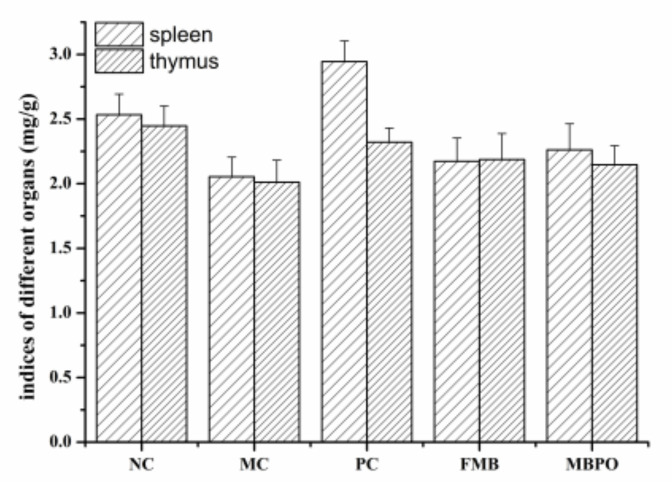
Index of spleen and thymus organs in mice.

**Figure 7 foods-11-02082-f007:**
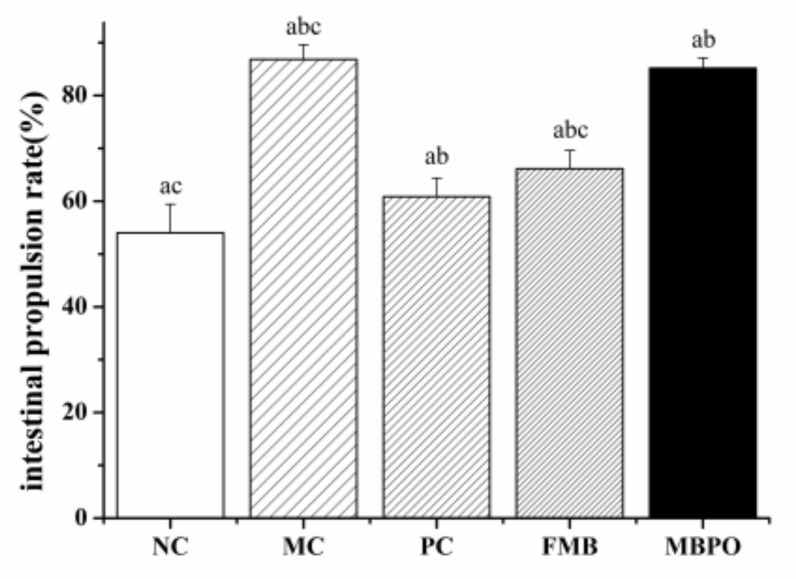
Intestinal mucosal SIgA content. ^a^
*p* < 0.05 vs. MC group. ^b^
*p* < 0.05 vs. NC group, ^c^
*p* < 0.05 vs. PC group.

**Figure 8 foods-11-02082-f008:**
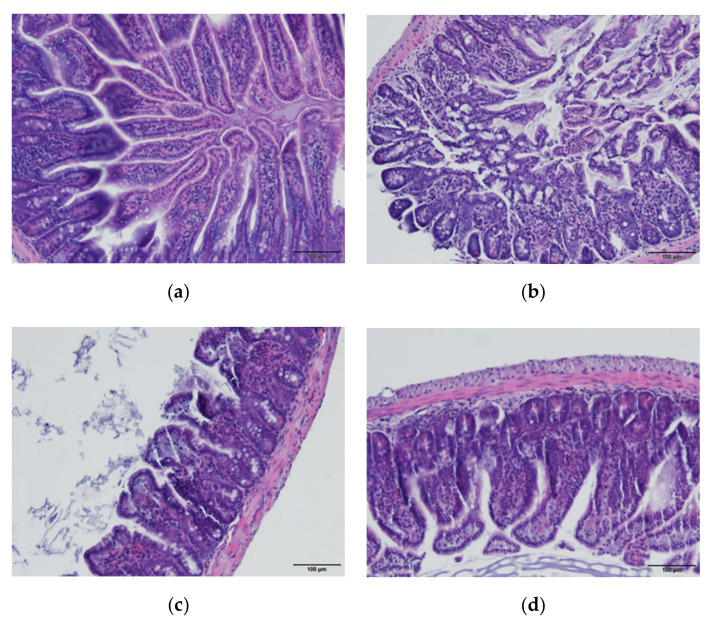
Representative images of H&E staining in colon sections of different treatment groups. (**a**) Normal control, (**b**) model control, (**c**) positive control, (**d**) fermented millet bran, (**e**) millet bran porridge oil.

**Table 1 foods-11-02082-t001:** Response surface test factors and level.

Level	Factors		
	A Inoculum Size/%	B Fermentation Time/h	C Fermentation Temperature/°C
−1	5	36	30
0	7	48	35
1	9	60	40

**Table 2 foods-11-02082-t002:** Design and results of the response surface experiment.

Test Number	A	B	C	Y_1_ Viable Count/ log CFU/mL	Y_2_ SDF Content/%
1	0	1	−1	7.48	9.92
2	−1	0	−1	7.35	9.16
3	0	1	1	7.65	10.76
4	1	−1	0	7.67	10.83
5	1	1	0	7.68	10.63
6	1	0	−1	7.79	10.42
7	−1	0	1	7.7	9.98
8	0	0	0	8.0	12.06
9	0	−1	−1	7.69	9.81
10	−1	−1	0	7.72	9.34
11	1	0	1	7.81	10.95
12	0	0	0	8.05	12.08
13	0	0	0	7.89	11.86
14	0	0	0	8.03	12.04
15	0	−1	1	7.87	10.48
16	0	0	0	8.06	12.17
17	−1	1	0	7.38	9.78

**Table 3 foods-11-02082-t003:** The result of the variance analysis.

Source	Sum of Squares		Degree of Freedom	Mean Square		F-Value		*p*-Value	
	Y_1_	Y_2_		Y_1_	Y_2_	Y_1_	Y_2_	Y_1_	Y_2_
Model	0.73	16.19	9	0.081	1.80	18.10	214.14	0.0005 **	<0.0001 **
A	0.080	2.61	1	0.080	2.61	17.79	310.76	0.0039 **	<0.0001 **
B	0.072	0.050	1	0.072	0.050	16.06	5.91	0.0051 **	0.0454 *
C	0.065	1.02	1	0.065	1.02	14.41	121.71	0.0067 **	<0.0001 **
AB	0.031	0.10	1	0.031	0.10	6.81	12.19	0.0349 *	0.0101 *
AC	0.027	0.021	1	0.027	0.021	6.06	2.50	0.0434 *	0.1577
BC	2.5 × 10^−5^	7.225 × 10^−3^	1	2.5 × 10^−5^	7.225 × 10^−3^	5.561 × 10^−3^	0.86	0.9426	0.3846
A^2^	0.17	4.26	1	0.17	4.26	38.12	507.24	0.0005 **	<0.0001 **
B^2^	0.15	3.34	1	0.15	3.34	34.44	397.90	0.0006 **	<0.0001 **
C^2^	0.085	3.48	1	0.085	3.48	18.82	413.69	0.0034 **	<0.0001 **
Residual	0.031	0.059	7	4.496 × 10^−3^	8.401 × 10^−3^				
Lack of fit	0.013	7.525 × 10^−3^	3	4.183 × 10^−3^	2.508 × 10^−3^	0.88	0.20	0.5211	0.8943
Pure error	0.019	0.051	4	4.730 × 10^−3^	0.013				
Cor total	0.76	16.25	16						

** means extremely significant (*p* < 0.01); * means significant (*p* < 0.05).

**Table 4 foods-11-02082-t004:** Effects of diarrhea in mice.

Group	Diarrhea Index		
	1st Day	4th Day	7th Day
NC	0	0	0
MC	2.04 ± 0.43 ^b^	1.67 ± 0.25 ^b^	0.84 ± 0.14 ^b^
PC	1.67 ± 0.24 ^b^	0.84 ± 0.15 ^ab^	0.21 ± 0.08 ^a^
FMB	1.70 ± 0.27 ^b^	1.28 ± 0.26 ^ab^	0.40 ± 0.11 ^ab^
MBPO	1.88 ± 0.18 ^b^	1.41 ± 0.22 ^b^	0.54 ± 0.14 ^ab^

^a^*p* < 0.05 vs. MC group, ^b^
*p* < 0.05 vs. NC group.

## Data Availability

Not applicable.

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
