# Peer review of "Antidiarrheal Effect of Fermented Millet Bran on Diarrhea Induced by Senna Leaf in Mice"

_foods, 2022, doi:10.3390/foods11142082_

Round 1
Reviewer 1 Report
I would like to provide my comments to the author as follows.
1. The author needs to take into consideration that the word "Bacillus natto" needs to be italic in the whole manuscript.
2. The English language and the grammar mistake are in the whole manuscript please take them into consideration.
3. What do you mean by this statement "were 10 used as indexes to study the effects of inoculum size, fermentation time and fermentation temper-11 ature on the fermentation effect," what is the fermentation effect??
4. what is the purpose of adding "it" in the following statement ? "and it also helps 38 to make millet bran a functional ingredient"
5. The ref. in the whole manuscript needs to be superscripted not written like numbers " in millet bran meal increased 8.88% 9"; "from 2.3% to 13.2% 12"; "intestinal metabolism 22, and enhance immunity 23"
6. Some ref. gives the following errors" intestinal microecological balance Error! Reference source not found" Please I do request the author to recheck the whole manuscript for errors from this form.
7. In section 2.3.1, what do you mean by this statement "After cooling, the suspension of 102 Bacillus natto was connected" specifically the word "connected"
8. The author needs to clarify what does he/she means by Table 1 "Codes and levels of response surface test" and the word response is written in correctly
9. Please do read again the "Anti-diarrhea Experiment of Fermented Millet Bran".
10. In section "2.6. Determination of Diarrhea Index in Mice" the ref. is still giving error
11. The resolution of all the figures needs to be improved.
12. In Table 2, the author needs to double-check it as there are mixed numbers and also the superscript is not correct
13. In line 393, Error! 393 Reference source not found..
14. The discussion part and the conclusion part need to rewritten again
15. I did not find a graphical abstract for the manuscript
Author Response
Dear Editors and Reviewers:
Thank you for your letter and for the reviewers’ comments concerning our manuscript entitled “Antidiarrheal Effect of Fermented Millet Bran on Diarrhea Induced by Senna Leaf in Mice” (ID: foods -1783763). Those comments are all valuable and very helpful for revising and improving our paper, as well as the important guiding sianificance to our researches. We have studied comments carefully and have made correction which we hope meet with approval. Revised portion are marked in red in the manuscript.
We tried our best to improve the manuscript and made some changes in the manuscript. We appreciate for Editors/Reviewers’ warm work earnestly, and hope that the correction will meet with approval. Once again, thank you very much for your comments and suggestions.
Yours Sincerely,
Minquan Hao
Corresponding author: Shujun Chen, Lizhen Zhang
Response to Reviewer 1 Comments
Point 1: The author needs to take into consideration that the word "Bacillus natto" needs to be italic in the whole manuscript.
Response 1: We accept your suggestion and have modified as required.
Point 2: The English language and the grammar mistake are in the whole manuscript please take them into consideration.
Response 2: We accept your suggestion and have corrected the English language and the grammar mistakes.
Point 3: What do you mean by this statement "were 10 used as indexes to study the effects of inoculum size, fermentation time and fermentation temper-11 ature on the fermentation effect," what is the fermentation effect?
Response 3: This statement means to the effects of different inoculation amount, fermentation temperature and fermentation time on viable bacteria count and soluble dietary fiber content of fermented rice bran were analyzed to determine the optimal inoculation amount, fermentation temperature and fermentation time.
Fermentation effect refers to the number of viable bacteria and soluble dietary fiber content in the fermented millet bran in the manuscript.
Point 4: What is the purpose of adding "it" in the following statement ? "and it also helps 38 to make millet bran a functional ingredient".
Response 4: "it" refers to the fermentation process mentioned in the previous sentence
Point 5: The ref. in the whole manuscript needs to be superscripted not written like numbers " in millet bran meal increased 8.88% 9"; "from 2.3% to 13.2% 12"; "intestinal metabolism 22, and enhance immunity 23".
Response 5: We accept your suggestion and have modified the format of references throughout the manuscript.
Point 6: Some ref. gives the following errors" intestinal microecological balance Error! Reference source not found" Please I do request the author to recheck the whole manuscript for errors from this form.
Response 6: We have rechecked the whole manuscript and corrected errors.
Point 7: In section 2.3.1, what do you mean by this statement "After cooling, the suspension of 102 Bacillus natto was connected" specifically the word "connected".
Response 7: The expression is wrong, it has been modified to “After cooling, the suspension of Bacillus natto was added to the culture medium and cultured in an incubator.”
Point 8: The author needs to clarify what does he/she means by Table 1 "Codes and levels of response surface test" and the word response is written in correctly
Response 8: The title of Table 1 has been changed to :“Response Surface test factors and Level”.
Point 9: Please do read again the "Anti-diarrhea Experiment of Fermented Millet Bran".
Response 9: We accept your suggestion and have revised the problems in the manuscript.
Point 10: In section "2.6. Determination of Diarrhea Index in Mice" the ref. is still giving error.
Response 10: We accept your suggestion and have corrected it.
Point 11: The resolution of all the figures needs to be improved.
Response 11: We accept your suggestion and have improved the resolution of all the figures to 600 dpi.
Point 12: In Table 2, the author needs to double-check it as there are mixed numbers and also the superscript is not correct.
Response 12: The above problems exist in Table 3, where we have rechecked the values and modified the superscript.
Point 13: In line 393, Error! 393 Reference source not found.
Response 13: We accept your suggestion and have corrected it.
Point 14: The discussion part and the conclusion part need to rewritten again.
Response 14: We accept your suggestion and have rewrittened the discussion part and conclusion.
Point 15: I did not find a graphical abstract for the manuscript
Response 15: We accept your suggestion and have added a graphic abstract for the manuscript at the end of the manuscript.

Reviewer 2 Report
Use Italic style for microorganisms name.
Row 137
............the other groups were given 1 g/mL senna decoction......
Do you analyse senna decocot composition? What about senosides?
I suggest to use Senna extract that is standardized.
Do you register water intake during the experiment?
Check the errors for references. Row 51
Author Response
Dear Editors and Reviewers:
Thank you for your letter and for the reviewers’ comments concerning our manuscript entitled “Antidiarrheal Effect of Fermented Millet Bran on Diarrhea Induced by Senna Leaf in Mice” (ID: foods -1783763). Those comments are all valuable and very helpful for revising and improving our paper, as well as the important guiding sianificance to our researches. We have studied comments carefully and have made correction which we hope meet with approval. Revised portion are marked in red in the manuscript.
We tried our best to improve the manuscript and made some changes in the manuscript. We appreciate for Editors/Reviewers’ warm work earnestly, and hope that the correction will meet with approval. Once again, thank you very much for your comments and suggestions.
Yours Sincerely,
Minquan Hao
Corresponding author: Shujun Chen, Lizhen Zhang
Response to Reviewer 2 Comments
Point 1: Use Italic style for microorganisms name.
Response 1: We accept your suggestion and have changed the name of microorganisms to italic style.
Point 2: Row 137 ............the other groups were given 1 g/mL senna decoction......Do you analyse senna decocot composition? What about senosides? I suggest to use Senna extract that is standardized.
Response 2: We did not analyze the composition of senna leaves, just refer to the method of preparing senna leaves in the literature and concentrate them to a certain extent. The specific preparation method is as follows: Exactly weigh 50 grams of Senna leaves, boil them in 300 mL of boiling distilled water for 20 min, then filter the residue, evaporate and concentrate the mixture of senna leaves to 1 g/mL, store it at 4℃, and heat it in the water bath at 25 ℃ when used.
Point 3: Do you register water intake during the experiment?
Response 3: The amount of water intake was not recorded during the experiment, only the amount of food intake.
Point 4: Check the errors for references. Row 51
Response 4: We have checked the errors references and corrected them.
